Comprehensive pathway-related genes signature for prognosis and recurrence of ovarian cancer

Zhao Xinnan 1
He Miao hemiao_cmu@163.com 2
1 Department of Rheumatology and Immunology, The First Affiliated Hospital of China Medical University , Shenyang , China
2 Department of Pharmacology, China Medical University , Shenyang , China
Uversky Vladimir
Electronic publication date: 2020 Dec 1
Publication date: 2020
Volume: 8
Electronic Location ID: e10437
Received 2020 Jul 13; Accepted 2020 Nov 6
Copyright: ©2020 Zhao and He
Copyright year: 2020
Copyright holder: Zhao and He
License: This is an open access article distributed under the terms of the Creative Commons Attribution License, which permits unrestricted use, distribution, reproduction and adaptation in any medium and for any purpose provided that it is properly attributed. For attribution, the original author(s), title, publication source (PeerJ) and either DOI or URL of the article must be cited.
License URL: https://creativecommons.org/licenses/by/4.0/

Keywords: Ovarian cancer, GSEA, LASSO, Overall survival, Recurrence-free survival

Funding: The authors received no funding for this work.

==============================
Background

Ovarian cancer (OC) is a highly malignant disease with a poor prognosis and high recurrence rate. At present, there is no accurate strategy to predict the prognosis and recurrence of OC. The aim of this study was to identify gene-based signatures to predict OC prognosis and recurrence.

Methods

mRNA expression profiles and corresponding clinical information regarding OC were collected from The Cancer Genome Atlas (TCGA) database. Gene set enrichment analysis (GSEA) and LASSO analysis were performed, and Kaplan–Meier curves, time-dependent ROC curves, and nomograms were constructed using R software and GraphPad Prism7.

Results

We first identified several key signalling pathways that affected ovarian tumorigenesis by GSEA. We then established a nine-gene-based signature for overall survival (OS) and a five-gene-based-signature for relapse-free survival (RFS) using LASSO Cox regression analysis of the TCGA dataset and validated the prognostic value of these signatures in independent GEO datasets. We also confirmed that these signatures were independent risk factors for OS and RFS by multivariate Cox analysis. Time-dependent ROC analysis showed that the AUC values for OS and RFS were 0.640, 0.663, 0.758, and 0.891, and 0.638, 0.722, 0.813, and 0.972 at 1, 3, 5, and 10 years, respectively. The results of the nomogram analysis demonstrated that combining two signatures with the TNM staging system and tumour status yielded better predictive ability.

Conclusion

In conclusion, the two-gene-based signatures established in this study may serve as novel and independent prognostic indicators for OS and RFS.

Introduction

Ovarian cancer (OC), one of the most common female malignant tumours, is the fifth-most common cause of mortality in women with severe gynaecological issues, with an estimated 13,940 deaths and 21,750 new cases reported in the 2020 United States cancer statistics (Siegel & Miller, 2020). Because OC is usually diagnosed at a late stage and currently lacks effective treatment options, patients with OC have extremely poor prognosis and only a 30–40% 5-year survival rate (Enroth et al., 2019). Despite the moderate improvements that have been made in the diagnosis and treatment of OC over the past 30 years, OC is still a highly malignant disease that is life-threatening (Li et al., 2019). In addition, OC patients have an average relapse-free survival (RFS) of two to three years after first-line therapy and die due to chemotherapy resistance (Dinh et al., 2008). To date, several common biomarkers for diagnosing OC remain ineffective due to lack of sensitivity and specificity; these include carbohydrate antigen 125 (CA125, AUC = 0.630) (Mansha, Gill & Thomson, 2019) and human epididymis protein 4 (HE4, AUC = 0.719) (Scaletta et al., 2017). At the same time, a number of studies have identified certain genes that are significantly associated with OC patient prognosis, such as TRIM44 (Wang et al., 2018) and CENPK (Lee et al., 2015). However, there is very limited prognostic value in a single candidate biomarker, which is attributed to inconsistent sample collection, detection methods, and small sample sizes. A number of studies have shown that combined biomarkers could improve prognostic accuracy compared to single biomarkers (Barata & Rini, 2017). Thus, extensive studies have attempted to establish molecular signatures based on gene expression to predict survival of patients, including mRNA- (Zhou et al., 2019), microRNA- (Li et al., 2019), and long non-coding RNA-based signatures (Xu et al., 2019). However, only a small number of prognostic signatures have been developed, and none have been directly applied in clinical practice. Hence, there is an urgent need to identify and validate novel, highly sensitive, and specific molecular biomarkers for prognosis, monitoring, and therapy improvement.

In this study, we sought to explore signalling pathways related to ovarian tumorigenesis and attempted to identify the potential predictive abilities of genes participating in these pathways for OC prognosis and recurrence. We established a nine-gene signature panel for overall survival (OS) and a five-gene signature group for RFS using LASSO Cox regression analysis. Both signatures have powerful prognostic predictive abilities and high sensitivity and specificity. Our results revealed that both signatures were independent risk factors for OC survival and recurrence. In addition, we constructed two nomogram models encompassing signature, TNM stage, and tumour status for OS and RFS, respectively, and found that both nomogram models exhibited high potential clinical utility by decision curve analysis (DCA). In general, we constructed two reliable and superior gene signature panels as prognostic predictors of OC patient survival and recurrence.

Materials and Methods

Data collection

The Cancer Genome Atlas (TCGA) –OC expression profiles (HTSeq-Counts) of genes and corresponding clinical information were downloaded from the TCGA database (http://cancergenome.nih.gov) (Tomczak, Czerwinska & Wiznerowicz, 2015). Detailed information is presented in Table S1. Other datasets, including GSE40595, GSE12470, GSE10971, GSE27651 GSE38666, and GSE17260, were downloaded from the GEO database (https://www.ncbi.nlm.nih.gov/geo/). Detailed information is shown in Tables S2–S3.

Gene set enrichment analysis

Gene set enrichment analysis (GSEA) (http://www.broadinstitute.org/gsea/index.jsp) was implemented using the JAVA.8 program using the Molecular Signatures Database (Subramanian et al., 2005). The hallmark gene sets were considered as a reference gene set and P < 0.05 was considered statistically significant.

Establishment of prognostic gene signatures for overall survival and relapse-free survival with LASSO Cox regression model

For gene sets in activated signalling pathways in OC, we first conducted univariate Cox regression analysis to identify genes significantly associated with OS and RFS in patients. Furthermore, LASSO Cox regression analysis was performed to construct optimal prognostic-predicting signatures for OS and RFS in patients by using the above candidate genes based on the “glmnet” R package. The LASSO penalty model successfully achieves shrinkage and selects variables simultaneously. The optimal values of the penalty parameter λ are determined through 10-fold cross-validations. Here, gene-based signatures for OS and RFS were established based on the expression of each prognostic gene and its coefficient, respectively. The risk score formula = β1 gene1 * expression of gene1 + β2 gene2 * expression of gene2+…βn gene n * expression of gene n, where β indicates the LASSO coefficients of these genes.

Gene signatures independent of other clinical characteristics

Univariate and multivariate Cox regression analyses were implemented to appraise the predictive abilities of gene signatures and clinical characteristics using SPSS v.22.0. The clinical characteristics in TCGA-OC included age, TNM stage, race, tumour status, and so on.

Clinical application of gene signatures

Nomograms can be used to predict the survival rates in diseases with multiple indicators (Iasonos et al., 2008). To determine the clinical application of gene signatures, we generated an optimal nomogram model using the “rms”, “nomogramEx” and “regplot” R packages to evaluate the 1-, 3- and 5-year OS and RFS of OC patients. DCA was performed to assess the clinical usefulness of the risk prediction nomogram (Vickers & Elkin, 2006). The x-axis represented the percentage of threshold probability, and the y-axis represented the net benefit.

Transcription factor analysis and GO enrichment analysis

Transcription factor analysis was produced using a website (https://www.gcbi.com.cn/gcanalyze/html/generadar/index). The clusterProfiler package was implemented to further explore the biological function of the transcription factor. P < 0.05 was considered a significant enrichment.

Statistical analysis

OS and RFS differences between patients in high-risk and low-risk groups were estimated by Kaplan–Meier survival curve analysis and calculated using log-rank tests. Time-dependent receiver operating characteristic (time-dependent ROC) curves were constructed to determine the sensitivity and specificity of gene signatures by calculating the area under curve (AUC) using the “survivalROC” R package (Do, Le & Le, 2020; Le, 2019; Le et al., 2019). We compared two groups using Student’s t-test for numerical variables and Chi-squared tests for categorical variables. All statistical graphs were produced using R (https://www.r-project.org/, v3.5.1), SPSS v.22.0 (SPSS Inc., Chicago, IL) and GraphPad Prism7 (GraphPad Software Inc., La Jolla, CA).

Results

Identification of signalling pathways correlated with OC tumorigenesis

To determine related signalling pathways involved in ovarian tumorigenesis, we performed GSEA on Hallmark gene sets for five independent GEO datasets, GSE40595, GSE12470, GSE10971, GSE27651, and GSE38666. The results showed that there are multiple signalling pathways significantly enriched in OC including “MYC_TARGETS_V1”, “MTORC1_SIGNALING”, “GLYCOLYSIS”, “E2F_TARGETS”, “G2M_CHECKPOINT”, “OXIDATIVE_PHOSPHORYLATION”, “DNA_REPAIR” and “PI3K_AKT_MTOR_SIGNALING” (Figs. 1A–1E and Table S4), suggesting that these signalling pathways might play critical roles in the development of OC.

Figure 1 The GSEA enrichment analysis of different GEO datasets.

(A) GSE12470. (B) GSE27651. (C) GSE38666. (D) GSE40595. (E) GSE10971.

Figure 2 The LASSO Cox regression model for OS and RFS of patients in the TCGA dataset.

The LASSO coefficients profiles of prognosis-related genes for OS (A) and RFS (C). Tuning parameter (λ) selection cross-validation error curve for OS (B) and RFS (D). The number of lines above the graph represent the number of genes in volved in the LASSO analysis, with the change of the Log lambda, the most representative genes were selected.

Establishment of gene signatures for overall survival and relapse-free survival

To evaluate the effect of genes in these pathways on OS and RFS, we performed univariate Cox regression analysis using TCGA data. The results showed that 48 and 44 genes were correlated with patient OS and RFS, respectively. Furthermore, the LASSO Cox selection method was implemented to establish a prognosis-predicting model (OS: Figs. 2A, 2B; RFS: Figs. 2C, 2D). Finally, a nine-gene signature panel for OS and a five-gene signature group for RFS were established by LASSO Cox regression analysis. The formula of the nine-gene signature panel for OS = (−0.0027) * expression of HMGB3 + 0.0122 * expression of PYGB + 0.0094 * expression of IDUA + 0.0092 * expression of LHX9 + (−0.0254) * expression of SLC7A11 + (− 0.0037) * expression of DPYSL4 + 0.0017 * expression of ANGPTL4 + (−0.0357) * expression of ISG20 + (−0.0353) * expression of GRM8; and the formula of the five-gene signature group for RFS = (−0.0114) * expression of SNRPA1 + (−0.0346) * expression of WARS + 0.0234 * expression of CITED2 + 0.002 * expression of ANGPTL4 + 0.0357 * expression of EFNA5 (Table S5). Next, we calculated risk scores for the OS and RFS of each patient based on the above formula. The distribution of risk scores, survival times, patient status, and expression heat maps of genes were obtained. The results showed that patients in the high-risk group had shorter OS and RFS and higher mortality than those in the low-risk group (Figs. 3A, 3B and Figs. S1A, S1B). According to these two signatures, patients were divided into high-risk or low-risk groups using the median risk score as the cut-off point (OS: median = 0.863; RFS: median = 0.457). We performed Kaplan–Meier analysis and found that patients with high risk scores had significantly shorter OS and RFS than those with low risk scores (OS: P <  0.0001, HR = 2.289, 95% CI [1.724–3.04]; RFS: P <  0.0001, HR = 1.783, 95% CI [1.465–2.264]; Figs. 4A, 4B). Furthermore, to explore the sensitivity and specificity of the risk score, we performed time-dependent ROC analysis. The results showed that for OS, the prognostic accuracy of the nine-gene signature panel was 0.640, 0.663, 0.758, and 0.891 at 1, 3, 5, and 10 years, respectively. The AUCs for RFS were 0.638, 0.722, 0.813, and 0.972 at 1, 3, 5, and 10 years, respectively (Figs. 4C, 4D). These results demonstrate that the nine-gene signature panel is a helpful indicator of OS for OC patients and that the five-gene signature group is a powerful prognostic predictor of OC patient recurrence.

Figure 3 Establishment of gene signature for OS and RFS in the TCGA dataset.

Risk score distribution, patients’ survival and status and expression heatmap of genes for OS (A) and RFS (B).

Figure 4 Kaplan–Meier curve analysis and Time-dependent ROC curve analysis.

Kaplan–Meier curve analysis for OS (A) and RFS (B). Time-dependent ROC curve analysis of the sensitivity and specificity for OS (C) and RFS (D).

Validation of the nine-gene signature panel for overall survival and five-gene signature group for relapse-free survival

To verify the prognostic value of our signatures, we investigated the independent dataset GSE17260 for validation of OS and RFS. We also performed Kaplan–Meier analysis and observed the same results as above, which showed that patients in the high-risk group had shorter OS and RFS and higher mortality than those in the low-risk group based on our nine- and five-gene-based signatures, respectively (Figs. 5A, 5B). In addition, we performed cross validation to prove the consistency of these signatures. We found the same gene sets in GSE17260 dataset (Figs. S2A, S2B), the only difference is the coefficients of each gene have changed a little. Furthermore, we also performed Kaplan–Meier analysis and observed the same results in TCGA dataset (Figs. S2C, S2D).

Figure 5 Kaplan–Meier curve analysis.

Kaplan-Meier curve analysis for OS (A) and RFS (B) in the GSE17260 dataset.

The signature perform better in survival prediction than other signature

Then, the ROC curves were implemented to contrast prognostic ability between our signature and other signature (risk score = 0.38*XPC −  0.24*PALB2 + 0.29*RECQL –  0.18*XRCC2 + 0.32*GTF2H5 –  0.19*GTF2H4 –  0.22*SSBP1 + 0.24*RAD54L –  0.25*MUTYH –0.3*SMUG1 –  0.16*TDP1 –  0.24*DDB2 + 0.26*RNH1 + 0.18*TP53BP1) (Sun et al., 2019). The result of ROC analysis illustrated that our signature has significantly higher ROC (AUC = 0.891) than other signature (AUC = 0.751) (Fig. S3). This result demonstrated that predictive power of our signature is greater than other signature. In general, our signature could be used as a superior indicator to predict the prognosis of OV patients.

Relationships involving gene signatures and clinical characteristics

To further investigate potential correlations between risk scores and patient clinical characteristics, we conducted Chi-squared tests. We found that for OS, the risk score was significantly associated with race and tumour status. Similarly, for RFS, the risk score was significantly correlated with TNM stage and tumour status. However, for both OS and RFS, there was no significant correlation between risk score and other clinical characteristics, including age and tumour grade (Tables 1 and 2).

Table 1 Correlation between risk score and clinicopathological features of OV patients for OS.

Characteristics	N	Risk score level	
		Low	High	P valuea	
Age(years)				0.518	
>65	114	60	54		
≤65	239	117	122		
TNM stage				0.534	
I and II	21	12	9		
III and IV	329	165	164		
Race				0.033	
Non-white	36	12	24		
White	307	160	147		
Tumor status				0.029	
Tumor free	79	48	31		
With tumor	228	106	122		
Grade				0.092	
G1+G2	42	16	26		
G3+G4	302	157	145		
Lymphatic invasion				0.721	
NO	43	22	21		
YES	94	45	49		
Notes.

Bold, significant values <  0.05.

a P values were calculated by X2 test.

Table 2 Correlation between risk score and clinicopathological features of OV patients for RFS.

Characteristics	N	Risk score level	
		Low	High	P valuea	
Age(years)				0.623	
>65	114	55	59		
≤65	239	122	117		
TNM stage				0.046	
I and II	21	15	6		
III and IV	329	161	168		
Race				0.738	
Non-white	36	19	17		
White	307	153	154		
Tumor status				0.002	
Tumor free	79	51	28		
With tumor	228	102	126		
Grade				0.175	
G1+G2	42	17	25		
G3+G4	302	156	146		
Lymphatic invasion				0.171	
NO	43	26	17		
YES	94	45	49		
Notes.

Bold, significant values < 0.05.

a P values were calculated by X2 test.

Prognostic value of gene signatures is independent of other clinical characteristics

To further evaluate the predictive ability of gene signatures and other clinical characteristics in terms of OS and RFS, univariate and multivariate Cox analyses were performed. As shown in Figs. 6A and 6B, the nine-gene signatures, stage, race, and tumour status were significantly correlated with OS, as seen in the univariate Cox regression analysis. Furthermore, multivariate Cox regression analysis results showed that the nine-gene signatures and tumour status were independent prognostic factors for OS. Meanwhile, univariate Cox regression analysis revealed that the five-gene signatures, staging, and tumour status were significantly associated with RFS. The five-gene signatures and tumour status were independent prognostic factors for RFS after multivariate Cox regression analysis.

Figure 6 Prognostic value of gene signature is independent of other clinical characteristics.

Forest plot summary of univariate and multivariate Cox regression analysis of gene signature and clinical characteristics on OS (A) and RFS (B).

Clinical application of gene signatures

We further constructed a nomogram predictive model combining gene signatures, race, stage, and tumour status to predict the probability of 1-, 3-, and 5-year OS and RFS in OC patients (Figs. 7A and 7C). The results showed that the nomogram model combined gene signatures, stage, and tumour status (OS: C-index: 0.743; RFS: C-index: 0.712). These results demonstrate that our gene signature-based nomogram approach exhibits superior performance in clinical settings. Furthermore, DCA was conducted to evaluate the clinical usefulness of the risk prediction nomogram for OS and RFS (Figs. 7B and 7D). The DCA results showed that for OS and RFS, the complex factor had a broader range of threshold probabilities and higher net benefits than single clinical characteristics or risk scores. Taken together, the DCA results demonstrate that our nomogram for OS and RFS has high potential clinical utility.

Figure 7 Nomogram to predict 1-, 3- and 5-year OS and RFS for ovarian cancer patients.

The nomogram for predicting survival rate of patients with 1-, 3- and 5-year OS (A) and RFS (C) (stage: 0 = stage I, 1 = stage II, 2 = stage III, 3 = stage IV; tumor status: 0 = tumor free, 1 = with tumor). Decision analysis curve for OS (B) and RFS (D).

Transcription factor analysis and GO enrichment analysis

Transcription factors are a family of DNA-binding proteins whose gene regulatory capabilities are of vital importance in defining the molecular state of a cell. Therefore, we performed Transcription factor analysis of genes above using website (https://www.gcbi.com.cn/gcanalyze/html/generadar/index). To explore the potential function of these transcription factor, we performed GO enrichment analysis by clusterprofiler package. The GO analysis results showed that these transcription factor analysis were mainly enriched in ”cell fate commitment”, ”pattern specification process”, ”embryonic organ development” and so on (Fig. S4).

Discussion

To a certain extent, indolent and aggressive cancers cannot be accurately stratified by the TNM staging system, which mostly relies on anatomical information without any molecular biological information or features (Choudhary et al., 2019; McIntyre et al., 2017). Therefore, it is indispensable to identify novel and effective predictive biomarkers for OC prognosis. In this study, we established and validated novel prognostic gene-based signatures for OS and RFS to improve the prediction of survival and tumour recurrence in OC patients.

Increasing number of studies are gradually clarifying the key signalling pathways involved in cancer initiation, progression, and metastasis, including the PI3K-Akt signalling pathway (Mabuchi et al., 2015), MYC signalling pathway (Dang, 2012) and so on. Moreover, the genes participating in these key signalling pathways might play critical roles in tumorigenesis. Therefore, it is necessary to explore the functions and roles of these key genes. Consistently, extensive studies have confirmed several signalling pathway-related genes that are significantly associated with the prognosis of cancer patients. Cao et al. (2019) confirmed that EMT-related genes promote bladder cancer subtype transition from non-muscle invasive bladder cancer (NMIBC) to muscle invasive bladder cancer (MIBC) and affect the prognosis of bladder cancer patients. In this study, our candidate genes for OS and RFS were screened for signalling pathways affecting the development of OC, including ”MYC_TARGETS_V1”, ”MTORC1_SIGNALING”, ”GLYCOLYSIS”, ”E2F_TARGETS” and ”G2M_CHECKPOINT”. MYC is one of the most highly amplified oncogenes in many human cancers and plays a pivotal role in tumorigenesis (Dang, 2012). Mammalian target of rapamycin complex 1 (mTORC1) signalling is found to be elevated in many human cancers, including OC, and promotes cell growth, proliferation, and metabolism (Menon & Manning, 2008). Increasing evidence suggests that aerobic glycolysis is related to tumour growth and chemotherapy resistance (Chakraborty et al., 2017). The E2F family plays a significant role in cancer initiation, progression, and resistance to therapy (Kent & Leone, 2019) and E2Fs can mediate fundamental cell-cycle deregulation in high-grade serous OC (DeMeyer et al., 2009). Hence, it is a plausible inference for these genes to be associated with the survival of patients with clear cell renal cell carcinoma (ccRCC).

High-dimensional data survival models have a high risk of overfitting, which is characterised by reduced importance when applying predictors to independent datasets. To solve this problem, we applied a Cox proportional hazard model using the LASSO penalty method to optimise gene selection. This is useful for selecting genes with strong prognostic value, high expression differences, and low correlation among each other (Tibshirani, 1997). The potential of LASSO has been previously implemented in colon cancer (Dai et al., 2018), bladder cancer (He et al., 2019) and other studies. Therefore, in this study, we successfully applied this method and established a nine-gene-based signature panel for OS and a five-gene-based signature group for RFS to optimise the predictive ability of prognosis in OC patients.

Although extensive studies have advocated several molecular biomarkers for risk stratification in OC, along with traditional clinicopathologic parameters, there are still some points to be improved as follows: (1) small sample sizes need to be enlarged; (2) independent validation datasets are needed to further confirm the stability and accuracy of the established signatures; (3) it is indispensable to compare the predictive power between signatures and the existing TNM staging system; and (4) it is more valuable to explore prognostic biomarkers for OC recurrence. In our study, we established novel superior prognostic signatures for OS and RFS in the TCGA dataset and validated the prognostic value of these signatures using an independent GEO dataset, suggesting the repeatability and practicability of our two signatures. Then, we confirmed that these signatures were independent risk factors for OS and RFS by univariate and multivariate Cox analyses. In addition, we constructed a nomogram model to better predict survival of OC patients and validated its promising clinical application by DCA. In general, our findings provided a superior and useful biomarker for OS and RFS in OC patients, based on the results of the LASSO regression analysis.

Our study demonstrated that nine-gene- and five-gene-based signatures for OS and RFS, respectively, were both closely associated with the prognosis of OC patients. However, the functions of most of the genes in our signatures have not been entirely elucidated. For genes associated with OS, the chromatin-associated high-mobility group box 3 (HMGB3) protein is upregulated and has poor prognosis, and its targeted depletion can attenuate cisplatin resistance in human OC cells (Mukherjee et al., 2019). PYGB has been shown to be upregulated in OC tissue and markedly associated with poor prognosis in OC patients (Zhou, Jin & Wang, 2019). LHX9 is a biomarker in ovarian stem cells (Auersperg, 2013). SLC7A11 is an independent prognostic risk factor for OS in OC (Yin et al., 2019). Low levels of DPYSL4 are significantly correlated with poor prognosis in OC (Nagano et al., 2018). In addition, Jiang et al. (2019) suggested that DPYSL4 is significantly associated with OS in hepatocellular carcinoma. High expression of ISG20 is related to poor clinical outcomes in glioblastoma (Gao et al., 2019). GRM8 is a tumour-suppressor gene in endometrial cancer cell lines (Liang et al., 2012). For genes associated with RFS, SNRPA1 is an oncogene in colorectal cancer (Zeng et al., 2019). Yanagie et al. (2009) demonstrated that CBP/p300-interacting transactivator (CITED2) is potentially a multidrug-resistance gene, which is consistent with our results. ANGPTL4 is a novel biomarker of hypoxia in OC, and is upregulated in partial responders to chemotherapy compared to responders (McEvoy et al., 2015). Yang et al. (2019) found that EFNA5 can be used as a novel prognostic biomarker and potential therapeutic target for OC patients.

Inevitably, there are certain innate limitations to our study that need to be addressed. First, screening for these signatures was based on data from TCGA and GEO datasets without validating their prognostic value in clinical practice. Second, although these candidate genes are involved in key signalling pathways that influence prognosis in OC, the mechanisms underlying some genes in our signature panels remains unclear. Undoubtedly, further investigation of the roles of these genes in OC progression is needed. It is reported that blacks vs whites were over two-fold less likely to receive a surgery-chemotherapy, suggesting that race may affect survival status of ovarian cancer patients greatly, which was consistent with our results. In addition, higher socioeconomic status is associated with a greater probability of undergoing surgical resection and with improved survival in patients with ovarian cancer, demonstrating socioeconomic status might also affect prognosis of patients. Unfortunately, several important clinical characteristics, such as treatment modality and socioeconomic status are not available in the TCGA cohort. Finally, prospective, large-scale, multicentre studies are indispensable to verify the prognostic ability of our signatures before applying them in clinical settings.

Conclusions

In conclusion, we established a nine-gene-based signature panel for OS, and a five-gene-based signature group for RFS using a LASSO Cox regression model. These signatures could be potentially novel and independent prognostic indicators for predicting survival and disease recurrence in OC patients. Moreover, combining these signatures with current clinical indicators may serve to optimise their predictive ability and guide future clinical planning for patient monitoring.

Supplemental Information

Supplemental Information 1 Risk score distribution in alive and dead patients for OS (A) and RFS (B)

Click here for additional data file.

Supplemental Information 2 Kaplan-Meier curve analysis for OS (A, C) and RFS (B, D) in the GSE17260 dataset and the TCGA dataset

Click here for additional data file.

Supplemental Information 3 Comparison of time-dependent ROC analysis of our signature with other mRNA signature

Click here for additional data file.

Supplemental Information 4 GO enrichment analysis of transcription factor

Click here for additional data file.

Supplemental Information 5 Clinical characteristics of OV in TCGA data set

Click here for additional data file.

Supplemental Information 6 Brief information of GEO datasets in the study

Click here for additional data file.

Supplemental Information 7 Clinical characteristics of OV in GSE17260 dataset

Click here for additional data file.

Supplemental Information 8 Summary analysis for Hallmark gene sets in GEO database

Click here for additional data file.

Supplemental Information 9 Pathways including LASSO genes

Click here for additional data file.

Additional Information and Declarations

Competing Interests

Author Contributions

Data Availability

The authors declare there are no competing interests.

Xinnan Zhao conceived and designed the experiments, performed the experiments, analyzed the data, prepared figures and/or tables, authored or reviewed drafts of the paper, and approved the final draft.

Miao He conceived and designed the experiments, analyzed the data, authored or reviewed drafts of the paper, and approved the final draft.

The following information was supplied regarding data availability:

The raw data are available from TCGA (Ovarian cancer) and GEO (GSE40595, GSE12470, GSE10971, GSE27651, GSE38666, and GSE17260).

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
