# Peer review of "Comprehensive pathway-related genes signature for prognosis and recurrence of ovarian cancer"

_PeerJ, doi:10.7717/peerj.10437_

## Round 0.1 · original submission · Major Revisions

Please address all the critiques of the reviewers and revise your manuscript accordingly.

Reviewer 1 ·

Basic reporting

NA

Experimental design

NA

Validity of the findings

NA

Additional comments

In this manuscript “Comprehensive pathway-related genes signature for prognosis and recurrence of ovarian cancer “ Zhao et al. have analyzed the gene expression profiles of ovarian cancer patients to develop a gene signature of overall survival and recurrence free survival using regression analysis. Although manuscript is written well but authors need to perform more rigorous data analysis to support their findings. Following are some of the comments that need to be addressed:

1. Figure 1: What does size on x- axis indicates? Are these p-values FDR corrected? How these barplots sorted ? Do authors find similar enriched pathways in TCGA data also?

2. What is the significance of using LASSO over ELasticNet which can also handle coexpression patterns of genes more efficiently.

3. Are risk scores significantly different in alive and dead patients? A boxplot representation of risk score in these two categories and statistically calculating the pvalue of significance can clearly show this instead of a scatter plot representation, which could be misleading.

4. What does authors want to represent in heatmap of Figure 3? Expression levels of genes shown in figure are not different in low and high risk groups. For a better representation gene expression should be normalized across the patient data.

5. How does cutoff between low and high risk identified? What is the overlap of genes in OS and RFS.

6. Figure 4C –D lacks legend. What does these color represent? It is not clear why red lines (risk scores) have so much variations in FPR and TPR values?

Reviewer 2 ·

Basic reporting

The article is written in a professional article structure. However, it contains unclear sentences due to grammatical errors, lack of formal language and precision, and concepts could be further explained. I would suggest that the authors review the manuscripts to provide precise examples and professional language.

Line 42 is missing a space before "(": 2020 cancer statistics (Siegel & Miller 2020)
Line 42-44: the sentence structure is unclear
Line 46 OC is still a highly malignant tumor which threat women's lives: should be rewritten in a professional manner, OC is a highly malignant tumor, "which threatens women's lives" should either be replaced with mortality statistics
Line 51: it is unclear why the current methods are unsatisfactory, no performance values arre provided
Line 54 identified some genes: "some" should be removed, either state the number of genes or just use examples
Line 57 "And": replace with a better connector such as furthermore
Line 65 "In this study, we were committed to explore": we explored (or investigated)
Line 66 "and attempted to identify the potential prognostic": say what you identified, remove "attempted".
Line 69, 70, and 72 say "both of the two": say "both"
Line 71 "Besides": use "In addition"
Line 120-122: fix the grammar and sentence structure as it is unclear
Line 122 "there are a lot of": state how many
Line 124-125 "and so on": either state them all or say "such as"
Line 128: it is not stated how many genes were tested
Line 151 "9-gene signature was a helpful overall survival's indicator of OV": this is not professional language and OV has not been defined
Line 164 "age, grade and so on": use professional language
Line 197 "studies has": studies have
Line colors used in the plots of figure 2 and 4 are not explained in the figure legend.
Figure 3 should show the color legend in risk score plots for A and B.
Figure 4 and 5 should show the hazard ratio and its 95% confidence interval

Experimental design

In this manuscript the authors use public available datasets to identify genes with prognostic value and tested their model. The research question is well presented. However, there are sections that should require other analysis.

Section 2.1: detailed clinical and demographic information is only provided for the TCGA samples, this information should be provided for all samples in order to asses the data that has been used in the study.

Section 2.3 and 2.6: it is not specified where the models were performed, was it R or other software?

Section 3.1: Why were only hallmark genes used for this study? Is it expected that only hallmark genes will have prognostic value? Further, the author should correct their p-values by multiple hypothesis testing.

Section 3.2: The correlation method and correlation threshold are not specified. Therefore, it is not possible to asses the quality of the genes for OS and RFS.
Line 132-133, it is not defined how this gene sets were defined.
Mean or median values should be provided for OS and RFS
Hazard ratios and their 95% confidence intervals should be shown in line 145.

Section 3.3
The authors use only one external dataset as validation for their prognostic markers. If other datasets are used to create the gene sets, are the same gene sets found? The authors should perform cross validation for their analysis to prove the consistency of their signal. Therefore the suggested experiments is to use a cross-validation using each of the datasets as validation while the others to identify the gene sets.

Section 3.5
Clinical and demographics data was only provided for one dataset. Was this the only dataset used for this analysis? In addition, identified race as a cofounder of the results should concern the authors. As race could imply socioeconomic differences in the patients and ultimately the difference in OS could be driven by these factors instead of the gene signatures.

In addition, the prognostic model in this study was not compared to other prognostics models on the same validation dataset. There we can't asses the quality, validity and improvement of this new prognostic method.

Validity of the findings

The conclusions of this paper are not supported by the results provided in the manuscript. The authors should use the cross-validation to test the robustness of their gene set and validation results. In addition, the correlation of race with OS and RFS is concerning, which may explain the difference in this parameters instead of the gene sets used. Further, the authors can't conclude that their model may be a superior prognostic indicator for OS and RFS if it is not compared to the other methods.

Additional comments

No further comments

Reviewer 3 ·

Basic reporting

The authors present a bioinformatic analysis for identifying genes signature for prognosis and recurrence of ovarian cancer. This is a well-known study and the authors proposed a new gene signature on it. Since the idea is of interest, there are major points that need to be addressed.

The authors should improve the English language usage in the paper, there contain many grammatical errors or typos such as:
- ... prognostic value in an independent GEO datasets.
- ... was the fifth major causes of mortality of women ...
- Despite the moderately improvement has been made in diagnosis and treatment of ...
- ...
They should re-check the errors and consider rewriting the content of the article in concise language.

Experimental design

The authors should provide detail on the data that they used (number of data, how many data for training, validation, ...).

Did the authors consider the inspection and removal of batch effects when merging different datasets from GEO?

Source codes should be provided for reproducing the results.

ROC curve & AUC were used in previous works in biomedical such as PMID: 31921391, PMID: 32613242, and PMID: 31362508. Therefore, the authors should refer more works in this description to attract broader readership.

Why did the authors use different datasets for different analyses, i.e., GEO in signaling pathways and TCGA in gene signature? In my opinion, the authors could use one set for generating the signatures and the other for validation.

Validity of the findings

Why was the performance from ROC curve analysis chaotic and inconsistent?

The authors should compare their gene signature with the previous works on ovarian cancer.

For this kind of problem, the authors should perform more analyses related to GO enrichment analysis and transcription factor analysis.

Additional comments

No comment.

---

## Round 0.2 · Minor Revisions

Please address remaining issues pointed by reviewer #2 and amend your manuscript accordingly.

Reviewer 2 ·

Basic reporting

The authors responded to my previous comments properly.

Experimental design

The authors responded to my previous comments properly.

Validity of the findings

Section 3.4 name is confusing. Also, include some metric value in the text to show how much better this new signature is
Section 3.6: is it independent considering the risk score association with other characteristics? The authors should add a socioeconomic part in their discussion. It is important to show the possible effect of race given that the study is retrospective and race could bias/cofound effects

Additional comments

No comments

Reviewer 3 ·

Basic reporting

No comment

Experimental design

No comment

Validity of the findings

No comment

Additional comments

My previous comments have been addressed well.

---

## Round 0.3 · accepted · Accept

Remaining issues were addressed and the revised manuscript is acceptable now.